# “Why Didn’t They Teach Us This?” A Qualitative Investigation of Pharmacist Stakeholder Perspectives of Business Management for Community Pharmacists

**DOI:** 10.3390/pharmacy11030098

**Published:** 2023-06-11

**Authors:** Braedon Davey, Daniel Lindsay, Justin Cousins, Beverley Glass

**Affiliations:** 1College of Medicine and Dentistry, James Cook University, Townsville, QLD 4811, Australia; beverley.glass@jcu.edu.au; 2School of Public Health, The University of Queensland, Brisbane, QLD 4072, Australia; d.lindsay@uq.edu.au; 3School of Pharmacy and Pharmacology, College of Health and Medicine, University of Tasmania, Hobart, TAS 7005, Australia; justin.cousins@utas.edu.au

**Keywords:** community pharmacy management, education, teaching, mentorship, leadership, culture, barriers, strategies

## Abstract

Expanding the scope of practice has provided an opportunity to reflect on the business management role of the community pharmacist. This study aimed to determine stakeholder perspectives of what business management skills are required for the community pharmacist, potential barriers impeding changes to management in the pharmacy program or community pharmacy setting, and strategies to improve the business management role of the profession. Purposively selected community pharmacists across two states in Australia were invited to participate in semi-structured phone interviews. A hybrid approach of inductive and deductive coding was used to transcribe and thematically analyse interviews. Twelve stakeholders described 35 business management skills in a community pharmacy, with 13 skills consistently used by participants. Thematic analysis revealed two barriers and two strategies to improve business management skills in both the pharmacy curriculum and community pharmacy. Strategies to improve business management across the profession include pharmacy programs covering recommended managerial content, learning from experience-based education and creation of a standardised mentorship program. There is an opportunity for business management culture change within the profession, and this may require community pharmacists developing a dual thinking process to appropriately balance professionalism and business management.

## 1. Introduction

Community pharmacists operate in a retail environment and require a balanced perspective between being a healthcare expert and achieving sustainable business management [1,2]. Operating in this retail setting has ingrained a ‘transactional model’ culture within the profession, where the image of the community pharmacist is framed as prioritising dispensing, compounding, and supply of medicines [3]. Emergence of the expanding scope of practice has provided an opportunity for the community pharmacist to shift their scope from being ‘retailers’ to that of healthcare professionals [2]. Various studies have described the expanding scope of practical roles that pharmacists can provide for their community, with findings such as improved health outcomes for patients, a reduction in costs, and reduced hospital admissions in chronic disease management, in a wide variety of healthcare settings [3,4,5,6]. Business management is important to the success and sustainability of an expanded scope of practice and, thus, the community pharmacy [7]. There is however a culture within the pharmacy profession that underappreciates the valuable skillset associated with business management [8]. Not only can business management improve the clinical role [9,10], but the opportunities presented from expanding the scope of practice requires individuals with strong management, finance, marketing, innovation, and entrepreneurship abilities to create new sustainable reimbursement models [7]. University educators in Australia [11], Canada [7,9], the United Kingdom [8], and the United Stated of America [12,13] are recognising the importance of educating pharmacy students in business, and evidence-based studies help inform their curricula on the business management skills to cover.

There have been developments in the business management field of the community pharmacist since the beginning of the 21st century. In 2002 Latif [14] identified there was a dearth of information available on the business management role for the community pharmacist and proposed a model for teaching effective managerial skills. Latif’s [13] contribution continued through the design of a management skills course for pharmacy students based on Katz’s [15] business management framework, which categorises management into conceptual, human, or technical skills. Latif identified that if managerial skills are important to the future success of pharmacy students, exploring how to appropriately include these skills in the curriculum should be investigated. He makes an important distinction between ‘teaching students about management’ and ‘students learning how to become managers’ [13]. This was subsequently supported by Perepelkin [9] who described that students could memorise and information-dump managerial content with little retention from one-way information exchanges, such as lecture-based teaching. Although Latif’s [13] course did not significantly improve pharmacy students’ managerial skills, it did confirm that business management skills are developed from experience, longer than one single semester, and individual character traits such as determination and persistence are important for success [16].

Not all community pharmacist business management investigations have employed the conceptual, human, and technical framework. The researchers are only aware of studies by Ram et al. [2] and Davey et al. [17] that utilised Latif’s recommendation [14]. Ram et al. [2] identified an important challenge for community pharmacist managers, describing an internal conflict between operating in a retail business and being a healthcare provider. There were similarities across studies by Ram et al. [2] and Davey et al. [17] which found business management to be an important role for the community pharmacist and human-category managerial skills a dominant skillset required in the profession. This is consistent with Katz’s framework; human-domain skills are required across all levels of management and represent 50% of the required managerial skills across all management roles [15].

Business management research is an important resource for pharmacy educational programs to make informed decisions about the management skills—whether they be technical, human, or conceptual—to be included in curricula [14], but these are limited in number. Current business management studies primarily explore the managerial skills to be included in the pharmacy curriculum [12,17,18] and how to best teach this material [9,13,19]. Generally, the findings of these studies recommend preparing students with business management skills during their tertiary education program and to approach this through non-traditional, multi-method, or innovative teaching, as these methods have positive learning outcomes. A 2012 study by Rollins et al. [20] investigated the exclusion of business management from the pharmacy curricula, which found that students may be disadvantaged from not receiving managerial training and concluded that pharmacy school curricula should sufficiently cover this subject to meet the Accreditation Council for Pharmacy Education standards [21,22]. More recently, Schuh’s [23] commentary argued that business management should be a significant element of pharmacy education.

With the evidence supporting the inclusion of business management education for pharmacists, this study aims to explore the perceptions of community pharmacist stakeholders of business management skills required for their role, the potential barriers to change that may exist to the inclusion of business management in the university pharmacy program and community pharmacy environment, and the strategies to overcome them.

## 2. Materials and Methods

### 2.1. Study Design

This investigation used a qualitative descriptive approach study design [24,25] focussing on the lived experiences of the participants, which were learned through semi-structured one-on-one phone interviews. The Consolidated Criteria for Reporting Qualitative Research (COREQ) was completed as a checklist to report this study (Appendix A).

### 2.2. Ethics Approval

This research was conducted under the auspices of James Cook University Ethics approval H8342 and University of Tasmania Ethics approval H0024440.

### 2.3. Australian Community Pharmacy Environment

The community pharmacy is generally considered a retail business in Australia. Regulations restrict community pharmacy ownership, with only a pharmacist permitted to own a community pharmacy in Australia. There are additional restrictions on the number of community pharmacies a single pharmacist can own, and this number varies across states. Pharmacists operate under varying business models and hierarchical management structures, including independent stores, corporate banner groups that share similarities to franchise groups, and private banner groups. Over the last 20 years, there has been a remarkable increase in the number of ‘big box model’ private banner group community pharmacies, which have dominated the market through advertising the lowest sale price of prescription medications to consumers [11].

### 2.4. Study Population

This study, informed by a scoping literature review [17] on business management skills required by the community pharmacist, employed semi-structured phone interviews with pharmacist stakeholders from Queensland and Tasmania in Australia. Pharmacist stakeholders were selected via purposive sampling to ensure that participants included a male, female, community pharmacist, community pharmacist locum, community pharmacist manager, and community pharmacist owner manager. Participants from different business models and geographical locations of community pharmacies were also purposively selected to include independent community pharmacies, banner group/corporate community pharmacies, and from pharmacies located in urban, rural, and remote areas. The Queensland and Tasmanian branches of the Pharmacy Guild of Australia emailed their respective community pharmacist members an invitation to participate in this study, with details of the study provided through an attached information sheet and a consent form provided.

### 2.5. Interviews

A semi-structured interview framework was scripted to explore pharmacist stakeholders’ perceptions surrounding business management roles, positives and/or challenges of business management in a community pharmacy, and the preferred content and delivery method for this content in university curricula. The structure of the interview was ‘funnel-shaped’, designed to deliver broad- to narrow-focussed questions [26]. To limit bias and gain consistency across interviews, a series of pre-written ‘probes’ and ‘cues’ were included in the interview framework to further explore pharmacists’ responses [27]. Validity of this study was achieved through investigators discussing the content and structure of the interview framework, until it was agreed that potential responses from participants would answer the study aims. A pilot phone interview with a volunteer community pharmacist was conducted to test the method and interview guide.

To keep a record of participants and ensure a diverse sample population of pharmacists was included, a table was created with category headings, pharmacist role, location, gender, experience, and business model. Participants returning a signed consent form via email were contacted and phone interviews conducted at the residence of the lead researcher if participants met the study population requirements at a mutually agreed time and audio recorded. At the completion of interviews, the audio recording was transcribed using Zoom [28], with the resulting data given a unique code name (Appendix B Table A1) to ensure data was deidentified.

### 2.6. Data Analysis

Proofed transcribed files were imported into NVivo (V12) [29] and a hybrid approach to data analysis was performed by the lead researcher guided by Braun and Clarke’s approach to thematic analysis [30]. Initial analysis involved a deductive approach, where pre-determined themes of interest were created into parent nodes: (1) extracted business management skills, (2) business management in the curriculum, and (3) business management in the community pharmacy. Each transcribed file was inductively coded to merge words and phrases into larger themes under each respective parent node.

Inductive coding of parent node (1) extracted business management skills described in each interview and categorised these descriptions into a business management skills framework [17]. This coding revealed the business management skills that were described to be used across a range of community pharmacist populations (Appendix B Table A1, Appendix C Table A2 and Table A3). Responses were explored between different populations of community pharmacists (Figure 1, Appendix C Table A3); location; (urban or rural and remote), business model of community pharmacy (independent, or banner group/corporate); and position of pharmacist (locum, pharmacist manager, owner/manager, or multi-store owner/manager).

## 3. Results

Twelve community pharmacist stakeholders participated in phone interviews (Appendix B Table A1) between October 2021 and February 2023. There were (*n* = 6) participants from Queensland, and (*n* = 6) participants from Tasmania, Australia. Phone interview duration ranged from 15 to 45 min and continued until theme saturation surrounding each topic was evident or pharmacists were unable to contribute further information.

### 3.1. Extracted Business Management Skills

Thematic analysis of stakeholder-described business management skills of daily duties, node (1), found that, across all participants, the pattern observed was 30 business management skills described by those from rural, remote, and urban areas; 29 business management skills described by those from independent and banner group/corporate community pharmacies; and 13 common business management skills described by all participants covering all categories of the community pharmacist (Figure 1). These skills were technical: business acumen, financial analysis, professional development, prior experience, and business model diversity; the following were human: confidence, being proactive, personnel management, self-awareness, and customer care; and the following were conceptual: inventory management, pharmacy operations, and general business management. When mapping out these 13 business management skills against Katz’s managerial framework [15], this skillset is proportionally representative of lower-level management. The only business management skill that was not described was entrepreneurship. Inductive coding and thematic analysis of parent nodes (2) and (3) revealed two common barriers and strategies across both nodes.

### 3.2. Barriers to Involving Business Management in the Pharmacy Curriculum

#### 3.2.1. Not Covering Business Management in the Pharmacy Programs

The absence of business management content in the pharmacy curriculum was consistently expressed by pharmacist stakeholders and highlights a key barrier to preparing community pharmacists for their managerial responsibilities. Some pharmacists described receiving university education in a limited number of managerial skills, which were generally finance-related and explored profit and loss statements. Human resources and staff management was a skillset described as required for the community pharmacist role and no stakeholder described receiving any university training in this managerial field. The general perception was university pharmacy programs were not covering business management with the justification that there is not enough time available in the curriculum due to the necessity of prioritising the clinical content.

“So much of the role is human resources these days…Definitely none of that taught at university”.(2-QLD)

“I didn’t learn anything about how to run the pharmacy and how to manage staff in university”.(1-TAS)

“Sometimes you work in pharmacy, and you think why did they not teach us how to do this [business management]”.(3-TAS)

“We didn’t get a single skerrick of business information, even just the basics”.(4-TAS)

“I think there is tight curriculum space and I think one of the things that tends to fall off is actual business management skills”.(5-TAS)

#### 3.2.2. Delivering Clinical-Work-Ready Pharmacists… Not Managers

Stakeholders believe university education should deliver ‘work-ready’ pharmacists; however, the definition of ‘work-ready’ varied. Most stakeholders felt ‘work-ready’ pharmacists would have basic business management competency, but there was the view that a purely clinically educated pharmacist was also considered ‘work-ready’. Stakeholders perceived the current level of university clinical education appropriate and that this content should not be reduced to cover business management. Pharmacists voiced concern that students are focussed on learning clinical skills at university and do not want to learn management, presenting another barrier to overcome.

“The problem is that not every pharmacist that goes to university is going to want to manage or want to end up in management”.(2-QLD)

“In a class setting you are just mainly thinking about patient interaction and your clinical knowledge”.(3-QLD)

“My perception on BPharm would be to come out clinically ready to be a health practitioner to support patients in their community… I find that the business skill aspect isn’t necessarily aligned with what the community is expecting of a community pharmacist”.(2-TAS)

### 3.3. Strategies to Improve Business Management in the Pharmacy Curriculum

#### 3.3.1. How Do We Prepare Students for Business Management?… That Is a Really Good Question

Stakeholders were consistent in their opinions that business management should be learnt from experience with guidance from mentors. They indicated a preference for workshops with guidance from pharmacist mentors, reflecting community pharmacy placements. Additionally, they cautioned that learning within a university setting could present a challenge for students to become competent with managerial skills if the main delivery method was lectures. The consensus was to teach management later in the course, but the amount of content to deliver varied from minimal to a full year. Stakeholders did believe it was important to create an accredited industry mentorship program to ensure students receive a minimum standard of guidance. There was an association observed across interviews, with stakeholders identifying their success in management as often the result of guidance from their mentor.

“It’s important to place them in a pharmacy where there is a feedback mechanism where that particular pharmacy has a good reputation for developing students. I think where they are placed, those proprietors should really be held to account with some sort of checklist, some sort of standard. They should respect that when they have students these guys are being moulded”.(1-QLD)

“It really comes from experience over the years and it also comes from having mentors, people that I either looked up to or learn from. I’ve always had a business coach to help me, I’ve always had somebody to help me”.(1-QLD)

“Mentoring… it would be really helpful if there was a core group of experienced pharmacists that wish to impart their experiences on younger people”.(5-TAS)

#### 3.3.2. Keep Business Management Simple

One perception from pharmacists was the business management content could be delivered with a simplified approach. In these stakeholders’ experience, managerial teaching can be over-complicated by exploring too much ‘nitty-gritty’ detail, and students fail to understand the basic purpose of why they are learning these skills. Approaching business management from a ‘keep it simple’ approach is a method that may help students grasp the basic concepts of the content they are learning.

“Teaching basic management concepts, how to problem solve, how to critically access situations, strategic direction all of those things that are not really taught”.(2-QLD)

“A full year in your final year of university set purely for business… where you have to learn the real basics of pharmacy business management”.(4-QLD)

“You often don’t need the nitty-gritty detail. What you really need is a ‘big hands, small maps’ type of thing”.(5-TAS)

### 3.4. Barriers to Involving Business Management in the Community Pharmacy

#### 3.4.1. Inconsistency in the Standards of Managerial Skills

Pharmacist stakeholders were consistent in responding that not all community pharmacists are either going to want or need all these managerial skills. This barrier was revealed from the mixed perceptions within the profession on what is considered to be the minimum management skills for a community pharmacist. Responses varied from very minimal to a diverse core set of skills.

“If you’re not managing a store there’s a whole skill set that’s less important… I mean they still overlap but, it’s less important”.(2-QLD)

“You have to have those basic [management] skills developed into you, similar to clinical skills, when you’re coming up”.(4-QLD)

“If people wish to pursue ownership, management, you know pharmacist in charge, with responsibilities for business administration, they’re skills that not everyone would necessarily need… I don’t think they are an essential requirement”.(2-TAS)

#### 3.4.2. Finding Time for Business Management

A common theme across pharmacist stakeholders was being time-poor due to a shortage of skilled staff, micro-managing, or the profession driving a high workload. This was more prominent in rural and remote areas but was also found in urban locations. A depleted workforce has resulted in a more challenging role for experienced business managers, and there is a developing trend of placing early career pharmacists into managerial positions without appropriate training. There were suggestions that the corporatisation of the community pharmacy business model, subtly re-educating patients into consumers, was increasing the challenges for the pharmacist to balance being a healthcare practitioner and manager.

“Short staffed everyday all day…just insanely busy and no good skilled staff… it’s hard to find good skilled staff”.(2-QLD)

“The reason there is a lack of employment and difficulty in maintaining pharmacists has exactly got to do with the environment that pharmacy is in right now”.(4-QLD)

“In a rural place they probably just want to try throw you straight in and get you going cause they’re desperate for staff… you just literally have to figure it out as you go”.(6-QLD)

### 3.5. Strategies to Improve Business Management in the Community Pharmacy

#### 3.5.1. We Need a Dual Thinking Process

Pharmacists described how finding the right balance between being a business manager and healthcare professional was a challenge for the community pharmacist. Some stakeholders have experienced a change within the profession since the introduction and dominance of corporate-model pharmacies, where patients are being viewed as consumers. Pharmacists described this as a difficult part of the role and emphasised the importance of education to achieve a balance between being a clinician and manager.

“They need to have this dual thinking process, always the professional with duty of care for the patient, but also being commercial”.(1-QLD)

“It’s good if you can have a balance between having some clinical work as well as management… after a while I really missed that patient contact”.(6-QLD)

“I think there is this disconnect, and I think a lot of pharmacists and perhaps young pharmacist proprietors have an opportunity to go either way, they can look at their clientele as being patients or consumers, they’ve lost the healthcare focus”.(5-TAS)

#### 3.5.2. Leadership and Mentorship Are Rewarding

Human resources and particularly personnel management were the most described business management roles. There were two distinct groups of responses, those who enjoyed supporting staff and those who found it the greatest challenge of management. Stakeholders who had received post-degree education, coaching, or mentorship in personnel management were likely to speak positively about the joy they received from being able to support people in their managerial role. They described the benefits of leadership and mentorship, such as improving culture, helping others with personal goals, and making a change in someone’s life.

“The actual role itself is really enjoyable, it’s nice just to be able to help people and help them do better”.(1-QLD)

“One of things I do like to do is be helpful to people… to see them succeed in a very competitive world brings me satisfaction and joy”.(5-TAS)

“The biggest positive you can have… is when you actually make a change in someone’s life, when you actually do that, that’s an incredible feeling”.(6-TAS)

## 4. Discussion

This study highlights that not all community pharmacists will be managers or want to be, creating a barrier to educating pharmacists on business management. The findings uncover a predicament for the pharmacy profession; managerial skills such as personnel management [31] are routinely used by community pharmacists and the perception is that all pharmacists at some point will be placed in a management role [14,32]. There is no standardised business manager blueprint used within the community pharmacist profession and this inconsistency in what are ‘core managerial skills’ was a barrier to knowing what skills to teach and how to teach them in the pharmacy program. Although stakeholders revealed barriers that universities may face to include business management in their curricula, the finding that almost all stakeholders described not receiving managerial training at all is an opportunity to implement change. Our finding that community pharmacists need to develop a dual thinking process—balanced between healthcare and business management—uncovers a common purpose of improving business management, a strategy that could be implemented through experience-based education programs with guidance from qualified mentors.

Community pharmacists operate under a variety of business models with different levels of corporate structure to deliver healthcare services. These changing business environments have a potential to influence the business management skillset required by a community pharmacist. Pharmacists in our study generally described business management roles as consistent across all business models of pharmacy, but the emphasis on particular managerial skills may change in urban and rural/remote locations. This perception was supported by the study by Davies et al. [8], where 84.9% of pharmacists responded that managerial skills are employed daily to perform their role and 79% of educators agreed that business management is being prioritised in the community pharmacy. All stakeholders in our study described using managerial skills during their routine workday, but we found a minority view, consistent with Davies et al. [8], that not all community pharmacists’ duties involved a managerial component. Community pharmacists routinely being required to perform managerial duties was our dominant finding, and this supports Fejzic and Barker’s [33] study on ‘work readiness’, where there was a shift in focus of the profession towards employing graduates with managerial skills and an ability to grow the business.

There is an opportunity to improve the business management culture within the profession. Stakeholders described the benefits of finding leaders within the profession to provide mentorship as a strategy to improve business management in both the pharmacy curriculum and the community pharmacy setting. This is not a new finding, with White [34] identifying in 2005 that there was a potential leadership crisis in pharmacies on the horizon and that mentorship is important for fostering our new leaders. This view was supported by Hawkins [35], who in 2010 investigated mentorship programs and the ideal characteristics of mentors. Stakeholders in our study did not link the importance of certain personality traits ideal for mentors, but they did suggest it was crucial to create a program that standardises mentorship within the workforce to ensure mentees receive a minimum level of care. What our study did find was that pharmacist mentors were time-poor, and therefore often could not provide adequate guidance for their mentees, resulting in early career pharmacists being disillusioned by being “dumped” in poor learning environments. Toxic business mentors can do more harm than good, and Hawkins [35] described similar findings such as being time-poor, micro-managing, ‘dumpers’ who are described as believing in the sink or swim approach, and destroyers of goals or the mentees purpose. It is not a difficult link to make between current business management practices in community pharmacies and some concerning outcomes within the profession, including suicide rates [36], burnout [37,38,39], work-related stress [40,41], dissatisfaction with management [42], and job dissatisfaction [43,44]. Participants in our study supported some of these findings, describing the difficulty in recruiting skilled staff, stress from being overworked, and being consequently time-poor. Without changes to business management education and strategies, this trend is likely to increase with further demands expected from this workforce [45,46,47]. Influencing change requires leadership, and there is evidence to prioritise the education and development of leadership within the profession to prevent a ‘leadership crisis’ [34,48,49]. Svensson et al. [50] provided insight into why this may be occurring, revealing that, despite recent reforms in pharmacy curricula, universities are still at risk of delivering students into the field that are followers and preserve the status quo [7,9].

University pharmacy programs are central in student business management education and, in the United States of America [32] and Australia [51], it is generally a requirement to cover this content in the curriculum. The majority of community pharmacists in this study believe it is important to include a core business management component in the pharmacy education program and offer an elective ‘advanced’ course. Stakeholders felt the opportunity to learn business management post university was missing and the creation of a post-degree course would benefit the profession. What managerial content to cover remains unclear from our investigation. The common stakeholder suggestion of including one core semester of business management is conflicting with Latif’s [13] findings where one semester was reported as insufficient for student improvement. However, stakeholders are not expecting students to enter the profession ‘work ready’ as business managers. What we found was stakeholders believe it is important for students to know the basics and learn the big picture. Pharmacy programs could aim to prepare students with an understanding of the basic components of a community pharmacist business management framework [17] to be competent and confident with lower-level business management skills as defined by Katz [15]. This would align with pharmacy stakeholders in this study who described 13 common business management skills used across a variety of community pharmacist roles (Figure 1), with these skills proportionally representative of lower-level management expectations (Appendix C Table A3). A barrier identified from participants in this study was the perceived lack of human resources education during their pharmacy university degree, and this was often described as the most challenging role. This finding was supported by Calomo [31], who recognised there is a lack of emphasis placed on human resources in the pharmacy field. Across Katz’s [15] three tiers of management, human-domain managerial skills represent 50% of the required skills in each tier.

University pharmacy educators across different countries have identified that there are challenges involved in successfully delivering a business management course [7,9,13,52]. Stakeholders in our findings identified barriers for pharmacy school educators including teaching business management to clinically focussed students, limited time available in an already tight curriculum, and the difficulty of learning managerial content in a classroom setting. These findings are consistent with educators delivering managerial content, who described similar challenges such as additional time demands [9,52], considerable investment in school resources [7], giving up ‘control’ of the class achieved through lecturing [13], long-term sustainability [52], and finding experience-based learning partners or experts to teach [7,53]. To improve learning outcomes, stakeholders suggested that experienced-based learning was important, with preference for workshops, mentorship, and learning in a community pharmacy setting. These stakeholder perceptions are consistent with studies [7,9,53] investigating innovative delivery methods of business management, which have resulted in improved student outcomes. Of importance to stakeholders was that this learning environment should be focussed on students developing a dual thinking process, achieving the right balance between being a healthcare professional and business manager. Stakeholders stressed the importance of having qualified mentors involved in the process of developing community pharmacists. Matching mentees with qualified mentors who possess leadership skills, who want to professionally develop, support, and nurture our future community pharmacists, is important.

Operating in a retail environment could be more challenging since the introduction of Hepler and Strand’s [54] pharmaceutical care model that shifted the focus to improving patient outcomes. Although stakeholders in this study did not use the term ‘identity conflict’, they described the importance of students learning how to think both from a healthcare and business perspective. Community pharmacists are clinically trained healthcare professionals, generally not trained to identify with the business management role [55], and this has potential to cause an identity conflict [56,57,58]. Some stakeholders in our study described how community pharmacists can struggle to know what the meaning of a community pharmacist is anymore, healthcare provider or retailer. Ram et al. [2] found this identity conflict in New Zealand; however, Perepelkin and Dobson’s [1] Canadian study reported low levels of conflict between professional and business roles. A potential reason for this low level of conflict was organisational structure, with a known relationship between organisational culture and employee satisfaction [59]; Latif [14] highlights the importance of organisation in effective management, and Canadian pharmacists [1] who reported a low level of conflict worked in well-defined organisational structures. Predominately, the stakeholders included in our study described routinely working as a solo community pharmacist for varying reasons. This lack of a hierarchical organisation structure, where the clinical and managerial responsibilities can fall upon a single community pharmacist, could be a potential barrier to improving the business management role due to an underlying identity conflict. In 2020, a study by Kellar et al. [60] stressed this point on the pharmacist’s clinical role; if pharmacists do not identify as clinicians, they do not see themselves in this role and therefore cannot “be” clinicians. Our findings suggest that it is important for the profession to create a hierarchical learning environment for business management, where there is a common purpose to transition the culture of the community pharmacist toward identifying their role as both a healthcare professional and business manager.

### Strengths and Limitations

A strength of this study is the inclusion of a diversity of community pharmacist roles across different states in Australia. There are limitations in our knowledge of the business management role of the community pharmacist. Although we made efforts to include a range of community pharmacist roles across two states in Australia, no community pharmacist described only being required to perform clinical duties in their role. This does potentially impact the generalisation of our findings and this is justified considering the small number of participants included in our study. It is important to note that pharmacists across both states spanned a large age range (>40 years), received different pharmacy educations, and were not always familiar with contemporary university curriculum content, and this may have influenced their perceptions. We did not explore the number of years of practice each pharmacist had in community pharmacy, or how much interaction there was with current pharmacy students or university pharmacy programs, all potential factors that could affect participant perspectives. Capturing a greater number of community pharmacists from more diverse roles, business structures, and different locations could generate different themes or further support our findings in this study and improve our understanding of business management for the community pharmacist.

## 5. Conclusions

Community pharmacists generally work in a retail environment and require a dual thinking process balanced between being a healthcare provider and business manager. This study has confirmed the importance of university pharmacy programs preparing students for management, but uncertainty remains about what content to cover and the amount of time the curriculum should dedicate to business management. There was a preference for a practical approach to learning business management through workshops, mentorship, and experience in a community pharmacy setting. Four barriers to business management change within the profession include pharmacy programs insufficiently covering managerial content, delivering clinically focussed ‘work ready’ pharmacists, time demands in a community pharmacy for effective management, and inconsistency surrounding the standard managerial requirements of a community pharmacist. Four strategies to improve business management include covering management in the pharmacy program, simplifying managerial content, developing pharmacists with a dual thinking process, and creating a standardised mentorship program. The overall finding, that there is a potential opportunity for a culture change to business management in the profession, directs future research towards exploring leadership in the community pharmacy.

## Figures and Tables

**Figure 1 pharmacy-11-00098-f001:**
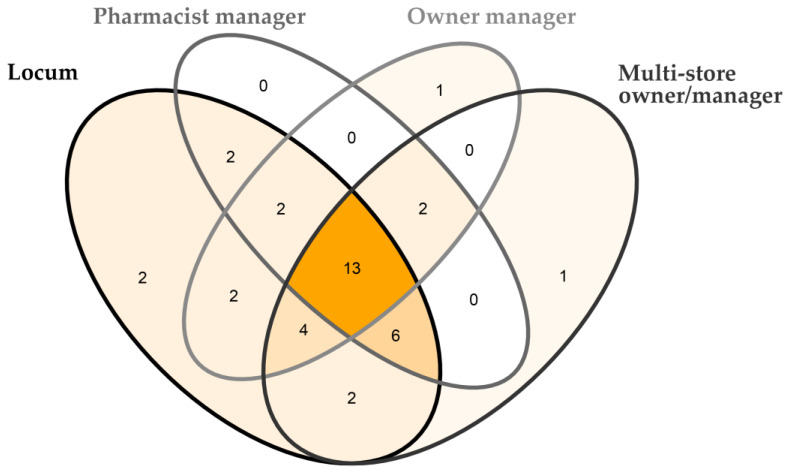
Venn diagram of the business management roles identified by community pharmacists, depending on their role displayed in 4 categories; pharmacist manger (pharmacist in charge, pharmacist manager), locum, owner manager, multi-store owner/manager (community pharmacist multi-store owner, community pharmacist multi-store manager). Numbers indicate the total business management skills described of the possible 38 as listed in Appendix C, Table A2 and Table A3.

## Data Availability

The research data is available in Figure 1, Appendix B Table A1, Appendix C Table A2 and Table A3. Individual reported results can be made available by request to the corresponding author.

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
