# Peer review of "“Why Didn’t They Teach Us This?” A Qualitative Investigation of Pharmacist Stakeholder Perspectives of Business Management for Community Pharmacists"

_pharmacy, 2023, doi:10.3390/pharmacy11030098_

Round 1

Reviewer 1 Report

Thank you for the opportunity to review your manuscript and research.  Overall, the paper was well presented and the conclusions fit the results.  I believe that there are a few areas for improvement, as outlined below:

1. line 27-29 within the abstract.  This statement seems out of place with the rest of the abstract. I think that you should rephrase the sentence to better align with the abstract results presented (there was no issues with alignment in the paper, but it did not align in the abstract).

2. lines 64-65 "exploring how to best for these skills to be included in the curriculum."- this statement does not sound correct.  I believe there are too many verbs.  I would suggest changing to "exploring how to appropriately include these skills in the curriculum."

3. Introduction- I felt that this entire section was too long and a bit redundant.  I feel that the paragraphs for lines 57-77 AND 78-98 should be more summarize, as both paragraphs are long and contain information that may not be necessary. 

4. Figure 1- While this was an interesting figure, I don't believe that it told me anything.  I would prefer to see either appendix A, table A1 Or a simplified Appendix B, Table 1.  I felt that both of these tables provided much more information that was relevant to the results compared to Figure 1. 

5. Lines 365 to 366 "and this was primary due to urban pharmacies routine business practice of discounting." - where is this information supported in either your results OR in the literature.  Also I think that you may need to clarify what this practice entails (as it could be many different practices from different viewpoints).

6. Section 4.1 Strengths and Limitations- I did not see any information about the participants interaction with current students OR how long ago they graduated from pharmacy school.  I feel that this should either be added to the paper OR addressed as a limitation, as interaction with schools of pharmacy may skew viewpoints about curriculum and curriculum needs. 

Author Response

Reviewer 1:

Thank you for the opportunity to review your manuscript and research.  Overall, the paper was well presented and the conclusions fit the results.  I believe that there are a few areas for improvement, as outlined below:

Response:

Thank you Reviewer 1 for taking the time to read the submitted manuscript and provide positive feedback. We appreciate your input to ensure this manuscript is a worthwhile addition to the literature.

  1. line 27-29 within the abstract.  This statement seems out of place with the rest of the abstract. I think that you should rephrase the sentence to better align with the abstract results presented (there was no issues with alignment in the paper, but it did not align in the abstract).

Response:

Yes, we agree with you and have edited our manuscript on line 25-30 within the abstract.

Strategies to improve business management across the profession include pharmacy programs covering recommended managerial content, learning from experienced-based education and creation of a standardised mentorship program. There is an opportunity for business management culture change within the profession, and this may require community pharmacists developing a dual thinking process to appropriately balance professionalism and business management.

  1. lines 64-65 "exploring how to best for these skills to be included in the curriculum."- this statement does not sound correct.  I believe there are too many verbs.  I would suggest changing to "exploring how to appropriately include these skills in the curriculum."

Response:

Yes, we agree with you and have edited our manuscript on line 62-64.

Latif identified that if managerial skills are important to the future success of pharmacy students, exploring how to appropriately include these skills in the curriculum should be investigated.

  1. Introduction- I felt that this entire section was too long and a bit redundant.  I feel that the paragraphs for lines 57-77 AND 78-98 should be more summarize, as both paragraphs are long and contain information that may not be necessary. 

Response:

Thank you and we agree. We have summarised some of the sentences and removed some of the information that may not be necessary for the reader. The changes in the manuscript have been made on line 56-72 and line 73-82.

There have been developments in the business management field of the community pharmacist since the beginning of the 21st century. In 2002 Latif[14] identified there was a dearth of information available on the business management role for the community pharmacist and proposed a model for teaching effective managerial skills. Latif’s[13] contribution continued through the design of a management skills course for pharmacy students based on Katz’s[15] business management framework, which categorises management into conceptual, human, or technical skills. Latif identified that if managerial skills are important to the future success of pharmacy students, exploring how to appropriately include these skills in the curriculum should be investigated. He makes an important distinction between ‘teaching students about management’ and ‘students learning how to become managers’[13]. This was subsequently supported by Perepelkin[9] who described that students could memorise and information dump managerial content with little retention from one-way information exchanges, such as lecture-based teaching. Although Latif’s[13] course did not significantly improve pharmacy students’ managerial skills, it did confirm that business management skills are developed from experience, longer than one-single-semester, and individual character traits such as determination and persistence are important for success[16].

Not all community pharmacist business management investigations have employed the conceptual, human, and technical framework. The researchers are only aware of studies by Ram et al.[2] and a Davey et al. [17] that utilised Latif’s recommendation [14]. Ram et al.[2] identified an important challenge for community pharmacist managers, describing an internal conflict between operating in a retail business and being a healthcare provider. There were similarities across Ram et al.[2] and Davey et al.[17] studies, finding business management an important role for the community pharmacist and human category managerial skills are a dominant skillset required in the profession. This is consistent with Katz’s framework; human domain skills are required across all levels of management and represent 50% of the required managerial skills across all management roles [15].

  1. Figure 1- While this was an interesting figure, I don't believe that it told me anything.  I would prefer to see either appendix A, table A1 Or a simplified Appendix B, Table 1.  I felt that both of these tables provided much more information that was relevant to the results compared to Figure 1. 

Response:

Thank you for this feedback. We agree that Table B1 in Appendix B provides more relevant information. However, we think that a simplified version of Table B1 will still be quite a large footprint in the main document and break-up the flow of the manuscript. To solve this, we created a Venn diagram, illustrating to readers that managerial skills are routinely used across the profession and there are numerous skills that overlap across different role types of the community pharmacist. Appendix A Table A1 is another good potential options, but we feel the Venn diagram is more justified in this location. 

  1. Lines 365 to 366 "and this was primary due to urban pharmacies routine business practice of discounting." - where is this information supported in either your results OR in the literature.  Also I think that you may need to clarify what this practice entails (as it could be many different practices from different viewpoints).

Response:

Thank you. On reflection we agree that we have not provided support for the statement “ due to the practice of discounting” in the results we have presented. We have removed this statement and the sentence on lines 358-360 In the manuscript now reads  as follows.

Pharmacists in our study generally described business management roles to be consistent across all business models of pharmacy, but the emphasis on particular managerial skills may change in urban and rural/remote locations.

  1. Section 4.1 Strengths and Limitations- I did not see any information about the participants interaction with current students OR how long ago they graduated from pharmacy school.  I feel that this should either be added to the paper OR addressed as a limitation, as interaction with schools of pharmacy may skew viewpoints about curriculum and curriculum needs.

Response:

Thank you, and we agree that this information should be covered in the strengths and limitations. We have included this in our manuscript line 476-482.

It is important to note that pharmacists across both states spanned a large age range (> 40 years), received different pharmacy education and were not always familiar with contemporary university curriculum content and this may have influenced their perceptions. We did not explore the number of years of practice each pharmacist had in community pharmacy, how much interaction there was with current pharmacy students or university pharmacy programs, all potential factors that could affect participant perspectives.

Reviewer 2 Report

This is a compelling topic of a manuscript given the current state of the pharmacy profession worldwide - thank you for your research and contributing your work! Overall this is a very worthwhile piece of exploratory research and starts an  important conversation within the pharmacy community regarding management, leadership, and entrepreneurship.  The research method while modest is appropriate and the findings are helpful.  I have no substantive comments or recommendations and hope this is simply the start of what will be additional scholarly contributions in this area from this group.

There is one specific area for improvement I would suggest:  readers unfamiliar with the context of Australian community pharmacy would benefit from more details regarding how community practice runs in that country: it is somewhat different than in other jurisdictions and this context directly influences your findings.  Further paragraphs detailing the structure and activities of community pharmacy (including HR and staffing issues, scopes of practice of personnel, funding of dispensing and clinical services, the profit-model/business model etc) would be helpful.

Author Response

Reviewer 2:

This is a compelling topic of a manuscript given the current state of the pharmacy profession worldwide - thank you for your research and contributing your work! Overall this is a very worthwhile piece of exploratory research and starts an  important conversation within the pharmacy community regarding management, leadership, and entrepreneurship.  The research method while modest is appropriate and the findings are helpful.  I have no substantive comments or recommendations and hope this is simply the start of what will be additional scholarly contributions in this area from this group.

Response:

Thank you Reviewer 2 for taking the time to read the submitted manuscript and provide positive feedback. We appreciate your kind comments and the recommendations you provided to improve our manuscript.

There is one specific area for improvement I would suggest:  readers unfamiliar with the context of Australian community pharmacy would benefit from more details regarding how community practice runs in that country: it is somewhat different than in other jurisdictions and this context directly influences your findings.  Further paragraphs detailing the structure and activities of community pharmacy (including HR and staffing issues, scopes of practice of personnel, funding of dispensing and clinical services, the profit-model/business model etc) would be helpful.

Response:

Thank you for this advice. We agree that providing context on how Australian community pharmacy practice operates would be beneficial to this paper. We think adding this content into the introduction may break the current  flow and make the introduction overly long. We have decided to include this information under the methods, section 2.3 Study population, which is line 113- 123 in our manuscript.

2.3 Australian community pharmacy environment

Community pharmacy is generally considered a retail business in Australia. Regulations restrict community pharmacy ownership, with only a pharmacist permitted to own a community pharmacy in Australia. There are additional restrictions on the number of community pharmacies a single pharmacist can own, and this number varies across states. Pharmacists operate under varying business models and hierarchical management structures, including independent stores, corporate banner groups that share similarities to franchise groups, and private banner groups. Over the last 20 years, there has been a remarkable increase in the number of ‘big box model’ private banner group community pharmacies, which have dominated the market through advertising the lowest selling price of prescription medications to consumers[11].

Reviewer 3 Report

An interesting and relevant paper. Great to see this addition to the literature.

Author Response

Dear Reviewer 3,

Thank you for reading our manuscript. We agree with you, and think this paper will be a worthwhile addition to the literature.